Gender differences in conference presentations: a consequence of self-selection?

Jones Therésa M. 1 theresa@unimelb.edu.au
Fanson Kerry V. 2
Lanfear Rob 3
Symonds Matthew R.E. 2
Higgie Megan 4
1 Department of Zoology, University of Melbourne , Australia
2 Centre for Integrative Ecology, School of Life and Environmental Sciences, Deakin University , Australia
3 Research School of Biology, Australian National University , Australia
4 Centre for Tropical Biodiversity and Climate Change, College of Marine and Environmental Sciences, James Cook University , Australia
Stewart Gavin
Electronic publication date: 2014 Oct 21
Publication date: 2014
Volume: 2
Electronic Location ID: e627
Received 2014 May 23; Accepted 2014 Sep 29
Copyright: © 2014 Jones et al.
Copyright year: 2014
Copyright holder: Jones et al.
License: This is an open access article distributed under the terms of the Creative Commons Attribution License, which permits unrestricted use, distribution, reproduction and adaptation in any medium and for any purpose provided that it is properly attributed. For attribution, the original author(s), title, publication source (PeerJ) and either DOI or URL of the article must be cited.
License URL: https://creativecommons.org/licenses/by/4.0/

Keywords: Gender and science, Women in science, Matilda effect, Conference presentations, Scientific visibility, Evolutionary biology, Leaky pipeline, Gender difference, Academic levels, Talk preference

Funding: There was no funding for this work.

==============================
Women continue to be under-represented in the sciences, with their representation declining at each progressive academic level. These differences persist despite long-running policies to ameliorate gender inequity. We compared gender differences in exposure and visibility at an evolutionary biology conference for attendees at two different academic levels: student and post-PhD academic. Despite there being almost exactly a 1:1 ratio of women and men attending the conference, we found that when considering only those who presented talks, women spoke for far less time than men of an equivalent academic level: on average student women presented for 23% less time than student men, and academic women presented for 17% less time than academic men. We conducted more detailed analyses to tease apart whether this gender difference was caused by decisions made by the attendees or through bias in evaluation of the abstracts. At both academic levels, women and men were equally likely to request a presentation. However, women were more likely than men to prefer a short talk, regardless of academic level. We discuss potential underlying reasons for this gender bias, and provide recommendations to avoid similar gender biases at future conferences.

Introduction

Gender discrepancies are present in many academic disciplines including science, engineering, the arts, humanities, and even gender studies, despite efforts to counter them (Hill, Corbett & St Rose, 2010; Kretschmer et al., 2012). To some extent, these differences are driven by perception biases and/or discrimination. For example, women are cited less often (Davenport & Snyder, 1995; Ferber & Brun, 2011; Maliniak, Powers & Walter, 2013), receive fewer awards and prizes (Lincoln et al., 2012), and have research that is valued less highly than men (Davenport & Snyder, 1995; Wenneras & Wold, 1997; Bornmann & Daniel, 2005; Knobloch-Westerwick, Glynn & Huge, 2013), even when relative opportunity is accounted for. In a recent experimental study involving students, men were evaluated as being more competent and worthy of a higher starting salary than women (Moss-Racusin et al., 2012). This suggests that perception biases may contribute to gender disparity from even the earliest stages of a career. However, gender discrepancies are also influenced by innate gender differences in behaviour. For example, women tend to publish less (Symonds et al., 2006; Conley & Stadmark, 2012; Larivière et al., 2013), use more tentative language (Leaper & Robnett, 2011), and ask for less when negotiating their salary (Tinsley et al., 2009).

Two metaphors are widely used to describe the current situation: (1) the ‘Matilda effect’, named after the American feminist critic Matilda Gage, who first described it, refers to the systematic under-recognition and denial of the contributions made by women in science, technology, engineering and mathematics (Rossiter, 1993; Knobloch-Westerwick, Glynn & Huge, 2013); and (2) the ‘leaky pipeline’, which describes the dramatic reduction in the proportion of women compared to men surviving each step up the academic (or equivalent) ladder (Pell, 1996; Winkler, Tucker & Smith, 1996). The underlying reasons for the presence of the Matilda effect and the leaky pipeline are diverse, complex and largely unresolved.

Identification and evaluation of gender discrepancies are often based on easily obtainable metrics, such as the number of publications, citations, or grants received. However, important dimensions of gender inequality may be overlooked by these assessments (West et al., 2012). One important metric, which may either directly or indirectly influence the perceived quality of a researcher, is their ‘visibility’ within their discipline or respective community (Damschen et al., 2005; Thelwall, Barjak & Kretschmer, 2006; Faulkner, 2009; Schroeder et al., 2013). In addition to publications and grant success, visibility may be achieved through conference attendance, presentations, plenary talks, and engagement with the media.

Recent studies present convincing evidence for consistent gender bias (favouring men) in terms of conference visibility. Using 21 years of annual meeting data for the American Association of Physical Anthropologists, Isbell and colleagues (2012) showed that men continue to dominate conference visibility, despite the fact that women comprise the majority of their membership. Men were more likely than women to request a talk in preference to a poster, and men requesting a talk were more likely than women to be allocated one (rather than a poster). Similarly, at the 2011 European Society for Evolutionary Biology congress (Schroeder et al., 2013), women were significantly under-represented among invited speakers, in part because they were more likely than men to turn down invitations to present. Studies have also shown that representation of women in symposia is positively related to the number of women on the organising committee (Isbell, Young & Harcourt, 2012; Casadevall & Handelsman, 2014).

These studies demonstrate that women currently have reduced exposure and visibility at conferences compared to men. Furthermore, they highlight the complex interplay between peer evaluation (Matilda effect) and a degree of active choice by women (either through their selection of a poster over an oral presentation, or by their declining an invitation to speak) that perpetuate these gender discrepancies. We note that the reasons for declining invitations to present are, in themselves, likely to be complex and diverse, and have been covered in detail by Schroeder et al. (2013). None of the above studies discriminated between academics (post-doctoral researchers and beyond) and students (Honours, Masters and PhD students). This distinction is important because gender-specific behaviour in conference participation may differ depending on academic level, and knowledge of such variation might be important in addressing gender bias.

Here, we use data from the 2013 Australasian Evolution Society (AES) conference, to assess gender bias in conference visibility at two different academic levels: student (Honours, Masters and PhD students) and academic (post-PhD). We first consider overall gender differences in conference visibility (i.e., the length of time men and women from the two academic levels spent presenting their research), and then use more detailed analyses to identify the source of any gender differences. The conference was unusual in two respects. First, it only offered oral presentations (short or long) and thus each presenter knew the amount of exposure they could expect a priori. Second, the sex-ratio of attendees was almost equal. Consequently, we should anticipate that on average women and men would receive roughly equal exposure. However, we found this was not the case. We narrow down the source of this gender difference, discuss the possible reasons causing it, and offer advice to future conference organisers as to how to encourage gender equality in visibility and exposure at their conferences. We hope that our study provides a useful template with which conference organisers and attendees can assess gender bias at future conferences. All authors attended this conference and two (MRES and KVF) were on the abstract selection committee.

Methods

Data collection

Four plenary speakers (two women, two men) accepted the committee’s invitation to speak at the 2013 Australasian Evolution Society (AES) conference. All other speakers were accepted through an open process of abstract submission. At the time abstracts were submitted, delegates who wished to present (N = 108/139; Table 1) were asked to (1) nominate whether they preferred to give a long talk (12 mins) or a short ‘speed’ talk (5 mins), and (2) specify whether they were a student (i.e., had not had their PhD conferred) for the purposes of student awards. A selection committee comprising three men and three women evaluated all requests and allocated them according to the provided abstract, with all delegates who requested a talk receiving one. However, conference time constraints resulted in a small number of delegates who requested a long talk being allocated a short talk (Table 1). The committee’s evaluations were not carried out blind to author(s) or career stage (student or academic). The gender of all delegates was identified post hoc (but prior to our analyses). No participant was identified in our study. Ethical approval was not required for the study as both the abstracts of all speakers (http://austevol.files.wordpress.com/2013/09/aes-2013-program.pdf) and the data relating to talk allocation (http://australasianevolutionsociety.com/~20talk-selection-aes2013/) are publicly available.

Table 1 Table of statistics for attendees to the AES conference.

Participation in talk presentations at the AES conference for the groups represented by student and academic women and men. The four invited plenary speakers (two women; two men) are not included.

Academic level	Gender	Attending	Presenting	Requested long talk	Received long talk	
			Yes	No	% Yes	Yes	No	% Yes	Yes	No	% Yes	
Student	Women	39	29	10	74%	18	11	62%	12	6	67%	
Men	27	24	3	89%	20	4	83%	18	2	90%	
Academic	Women	31	26	5	84%	16	10	62%	14	2	88%	
Men	42	29	13	69%	26	3	90%	23	3	88%	

Analytical approach

The conference delegates were categorised in four groups: student women, student men, academic women, and academic men. ‘Students’ were defined as Honours, Masters or PhD students, while ‘academics’ were defined as postdoctoral research associates, research fellows, and all other academics. For each of the statistical analyses (described below) differences among the four groups were assessed for each metric. Planned contrasts to directly compare student women and men, and academic women and men, were carried out for each metric as women and men at the same academic level should be comparable. The four plenary speakers were excluded from all analyses as they did not have to decide what type of presentation to request, and were not subject to the same selection procedures as other delegates. All statistical analyses were performed in SAS version 9.3 for Windows 7 (64-bit).

We conducted five sets of analyses. First, we analysed whether there was any significant difference from equal attendance by the four groups at the conference. This statistical test does not tell us whether there is equal participation by each group because we do not know the true distribution of Australasian evolutionary biologists in each of these groups. However, it does indicate whether attendees walking around would have gained the “impression” that they were observing each group evenly. To assess whether there was equal attendance among groups at the conference, attendance ratio was tested in the SAS procedure Freq using a chi-square test for equal proportions, with exact p-values specified. Planned contrasts directly comparing student men and women, and academic men and women, were carried out in SAS procedure Freq by testing for equal proportions in conference attendance using the binomial test. Two-sided p-values were selected as we did not a priori predict that conference attendance would be higher in one gender.

Second, we examined whether there was a difference among the four groups in ‘exposure’ of their research, as measured by the average length of time each attendee spent in front of the audience presenting their scientific research. We assigned an attendee who did not present the exposure value ‘0 min’, short-talk presenters ‘5 min’, and long-talk presenters ‘12 min’ presenting their research. Analysis of ‘exposure’ was carried out using two sets of data: ‘exposure—all attendees’, which were all 139 attendees including those who did not present (i.e., including attendees who were assigned 0 mins); and ‘exposure—presenters only’, which only included the 108 attendees who presented (i.e., excluding attendees assigned 0 mins). The ‘exposure—all attendees’ metric is an attempt to coarsely represent the overall and general impression of each group to other conference attendees. In contrast, the ‘exposure—presenters only’ analysis may be more representative of the impression given to conference attendees who only observed talks, without observing the ratios of each of these groups sitting around them in the audience or during breaks; or to those conference attendees who entirely weighted their impression of each group based on the presenters they saw, ignoring interactions they had when outside the presentation sessions.

Analysis of both ‘exposure’ metrics was carried out using a Kruskal–Wallis non-parametric one-way analysis of variance (ANOVA) using the SAS procedure Npar1way with the ‘Wilcoxon’ option specified, where the response variable was ‘exposure’, and the independent variable was ‘group’ (four categories as above). A non-parametric ANOVA was used because the response variable ‘exposure’ can only have one of two or three values (0, 5, and 12 min), therefore the data is not normally-distributed. The two-sample Kruskal–Wallis test statistic produces p-values that are equivalent to two-sided p-values from a Wilcoxon two-sample rank-sum test. Planned contrasts were used to directly compare ‘exposure’ for women and men within each academic level.

Finally, we attempted to find the basis of any significant differences in ‘exposure’ by analysing three metrics that reflected both the decisions of the attendees and any bias for or against each group: preference to present a talk or not; of those choosing to give a talk, preference for a long talk over a short talk; and, of those requesting a long talk, likelihood of being assigned a long talk rather than a short talk. The first two metrics address the differences in preferences for conference participation among the four groups, while the final question helps to address any potential bias for or against the four groups.

For these three metrics, a likelihood ratio test in SAS procedure Logistic was used to assess for significant differences among groups by comparing the logit model with and without the independent term ‘group’ (Agresti, 2013). To assess whether there was a difference among groups in their decision to present a talk (i.e., whether they submitted an abstract), the response variable was ‘present talk’ (categorical: ‘yes’ or ‘no’). To test whether there was a difference among groups in their preference for a long talk over a short talk, only the subset of attendees who chose to present a talk was analysed and the response variable was ‘preference for long talk’ (categorical: ‘yes’ or ‘no’). Finally, to assess whether there was a difference among groups in the likelihood of receiving a long talk when requested, only the subset of attendees who requested a long talk in their conference registration was analysed and the response variable was ‘received long talk’ (categorical: ‘yes’ or ‘no’). For all analyses, planned contrasts were also carried out by restricting the data to directly examine gender differences within each academic level (student and academic).

Results

Is there a difference among groups in attendance?

There was no significant difference in attendance across the four groups (exact Chi-square test for equal proportions: χ2 = 4.17, df = 3, exact P = 0.248; Table 1). Furthermore, student men and women did not differ in their attendance ratio at the conference (binomial test for equal proportions: z = 1.48, df = 1, two-tailed P = 0.140), and nor did academic women and men (binomial test for equal proportions: z = −1.29, df = 1, two-tailed P = 0.198).

Is there a difference among groups in the amount of exposure to their colleagues?

When considering all conference attendees, including those who did not present, there was no significant difference in the amount of time the four groups spent presenting their research to their colleagues (Kruskal–Wallis one-way analysis of variance (ANOVA) on ‘exposure—all attendees’: χ2 = 7.03, df = 3, P = 0.071; Fig. 1A), although it came close to significance. However, when considering the two academic levels separately, student women spent a highly significantly shorter period of time (36% shorter: average of 5.9 mins vs. 9.1 mins) presenting their research to their colleagues than did student men (Kruskal–Wallis one-way ANOVA on ‘exposure—all attendees’: χ2 = 7.44, df = 1, P = 0.006; Fig. 1A). On the other hand, academic men and women did not differ in the amount of time (average of 7.4 mins vs. 7.3 mins) spent presenting their research (Kruskal–Wallis one-way ANOVA on ‘exposure—all attendees’: χ2 < 0.01, df = 1, P = 0.995; Fig. 1A).

Figure 1 Time spent presenting at the AES conference.

Average time (minutes) spent presenting scientific research (‘exposure’) for student and academic women and men at the AES conference. (A) Exposure of all attendees. This average also includes attendees who did not present a talk and so may reflect an impression of an observer who attends and weights their impression by all aspects of the conference. (B) Exposure of presenters only. This average includes only those attendees who presented a talk and so may reflect the impression of an observer who only observes the talk presenters and gauges no other impression from the audience around them, or else weights their impression of each group solely based on presenters. Significance values of planned contrasts: ∗∗P < 0.05; ∗∗∗P < 0.01.

Differences in ‘exposure’ among groups were exacerbated when considering only those delegates who chose to present, with a significant difference among the groups in presentation time (Kruskal–Wallis one-way ANOVA on ‘exposure–presenters only’: χ2 = 11.28, df = 3, P = 0.010; Fig. 1B). These differences among groups occurred at both the student and academic level, and in the same direction. Of the students presenting, student women spent significantly less time (23% shorter: average of 7.9 mins vs. 10.3 mins) presenting their research than student men (Kruskal–Wallis one-way ANOVA on ‘exposure—presenters only’: χ2 = 5.93, df = 1, P = 0.015, Fig. 1B). Of the academics presenting, academic women also spent significantly less time (17% shorter: average of 8.8 mins vs. 10.6 mins) presenting their research than academic men (Kruskal–Wallis one-way ANOVA on ‘exposure–presenters only’: χ2 = 3.96, df = 1, P = 0.047, Fig. 1B).

Is there a difference among groups in preference for presenting a talk?

There was no significant difference across the four groups in preference for presenting a talk (likelihood ratio (LR) test for logit model: χ2 = 4.93, df = 3, P = 0.177; Table 1; Fig. 2A), and equivalent patterns were found when directly comparing within academic levels (LR test for logit model: women vs. men students: χ2 = 2.25, df = 1, P = 0.133; women vs. men academics: χ2 = 2.18, df = 1, P = 0.140; Fig. 2A).

Figure 2 Participation at the AES conference.

Comparing participation of student and academic women and men at the AES conference. (A) Percentage choosing to present a talk. (B) Of those choosing to present a talk, percentage who prefer a long talk over a short talk. (C) Of those who prefer a long talk over a short talk, percentage of those who were assigned a long talk. Significance values of planned contrasts: ∗P < 0.1; ∗∗P < 0.05.

Given the lack of difference among groups in preference for presenting a talk, the difference in exposure between genders seen above is predominantly due to giving a long vs. a short talk. Only 41% (12/29) of presenting women students gave a long talk, as opposed to 75% (18/24) of presenting men students. Similarly, only 54% (14/26) of presenting academic women gave a long talk, compared to 79% (23/29) of academic men. Below we tease apart what caused these differences between the genders in presenting long talks.

Of those choosing to give a talk, is there a difference among groups in preference for a long talk over a short talk?

Of those presenting a talk, there was a significant difference among groups in their preference for a long talk over a short talk (LR test for logit model: χ2 = 9.55, df = 3, P = 0.023; Table 1; Fig. 2B). Student men tended to have a greater preference for a long talk over a short talk than student women, although the difference was not significant and is based on a relatively low sample size (LR test for logit model: χ2 = 3.03, df = 1, P = 0.082; Fig. 2B). At the academic level this is shown much more clearly: academic men were significantly more likely to prefer a long talk over a short talk compared to academic women (LR test for logit model: χ2 = 6.22, df = 1, P = 0.013; Fig. 2B).

Of those preferring a long talk over a short talk, is there a difference among groups in likelihood of being assigned a long talk?

Of the attendees who preferred a long talk, there was no significant difference among the four groups in the likelihood of the conference organisation committee assigning them a long talk (LR test for logit model: χ2 = 4.44, df = 3, P = 0.218; Table 1; Fig. 2C). However, student women tended to be less likely than student men to be assigned a long talk when requested, although this difference was not quite significant and is based on a relatively low sample size (LR test for logit model: χ2 = 3.20, df = 1, P = 0.074; Fig. 2C). There was no significant difference between academic women and men in the likelihood of being assigned a long talk when requested (LR test for logit model: χ2 < 0.01, df = 1, P = 0.926; Fig. 2C).

Relative importance of factors contributing to decreased exposure for women presenters

Above, we found that student women who chose to present had 23% less time presenting than student men. No single factor was responsible, with both weaker preference for a long talk and allocation bias of a long talk being marginally non-significant contributors. If student women had the same preference for a long talk as student men (but allocation for long talks did not change), student women presenters would have 13.3% less time presenting than student men (as opposed to the 23% observed). An almost identical 13.1% difference in women student presenting time would be achieved if allocation for long talks was equal to men students (but preference for long talks did not change). This highlights that for women students, both talk preference and talk allocation contribute equally to their reduced conference visibility as compared to men students.

In contrast, the 17% less time spent presenting by academic women compared to academic men was almost solely due to a single factor: preference for a long talk. If academic women had the same preference as academic men for a long talk (but allocation for long talks did not change), academic women presenters would only have 0.6% less presentation time than academic men (as opposed to the 17% observed). If allocation for long talks for academic women was equal to academic men (but preference for long talks does not change) they would still be presenting for 16.5% less time.

Discussion

We found that women spent significantly less time presenting their research at the 2013 Australasian Evolution Society conference compared to men. This discrepancy was driven primarily by a stronger preference by men (or a weaker preference by women) for long talks. Our results highlight that even marginal differences in the presentation strategy used by women and men result in significantly different outcomes for exposure and visibility. Our results correspond with previous research exploring gender differences in conference presentations decisions (Isbell, Young & Harcourt, 2012), which showed men had a stronger preference than women for presenting a talk over a poster.

These data raise two interesting questions: (1) why do women and men have apparently different presentation strategies, and (2) does this difference in presentation approach affect a scientist’s visibility and perception by colleagues (i.e., do these differences matter)? These questions are pertinent to the broader issue of gender discrepancies in science, and they will be important to address as we move toward resolving the multitude of underlying reasons promoting them.

Gender differences are apparent in a range of academic tasks. Women are more likely to use tentative language when presenting their research (Leaper & Robnett, 2011), reject invitations to speak (Schroeder et al., 2013), prioritize teaching over research (Winslow, 2010), are less likely to cite their own work (Maliniak, Powers & Walter, 2013) or have a webpage (Barjak, 2006) when compared to men of equal career stage. Each of these approaches or strategies may result in reduced visibility for women scientists, and ultimately may contribute to and exacerbate the pre-existing gender imbalance.

The overall gender balance at the Australasian Evolution Society conference (2013) in terms of attendance and participation in talks for both students and academics was not significantly different from equal (c.f. Schroeder et al., 2013). This is encouraging for the future of evolutionary biology as a field and is reflective of an academic society that is acutely aware of the need to promote gender balance. What is startling is that, while delegates at the conference would have encountered equal numbers of women and men delegates, there remained a consistent gender bias in the nature of talk preference, which led to an overall difference in the exposure and visibility of women and men presenters (sensu Schroeder et al., 2013). Student women engaged in significantly less time presenting their research to colleagues than student men (Fig. 1). This amounted to 23 or 36% (equating to approximately 2 or 3 min on average) less exposure on average for student women compared to student men, depending on whether you considered only those who presented or student attendees overall, respectively. Although the number of academic women attending did not differ overall from academic men, when considering only those who presented, academic women as a population also spent significantly less time presenting their work (Fig. 1B): on average academic women presenters spent 17% less time presenting their research than academic men.

We found that these differences in time spent speaking predominantly arose as a result of gender bias in preference for long talks over short talks. Student and academic men had a significantly higher preference for long talks than women presenters of the same level (Fig. 2B), with a greater percentage of women choosing to present short ‘speed’ talks than their equivalent men colleagues. By their very nature, short talks allow less detailed and comprehensive presentation of scientific research than long talks. Therefore, by choosing to present short talks women presenters may be portraying their scientific research and skill sets less comprehensively than men at an equivalent stage.

Why were women less likely than men to request a long talk? That the patterns were almost identical between student and academic women suggests that there is some inherent gender difference and that the observed patterns are not influenced by which women survive the leaky pipeline. We offer three reasons for the potential differences in presentation choices. First, it is conceivable that the student and academic women were on average more junior than men in the same category. Junior students would be more likely to present preliminary research or their proposed research plan, and therefore be likely to choose (and be advised to choose) a short talk. This is compared to students in their final year of PhD research, who would be presenting long and comprehensive talks to cover their research outcomes, and in order to increase their chances of securing post-doctoral research opportunities in the near future. Similarly, junior academics, especially those on short-term research contracts, may be more likely to present short talks as they are more likely than established researchers to have recently changed jobs and/or research area and so have preliminary data more suited to a short talk. A coarse examination of our data suggested that, within the academics, the patterns were comparable for postdoctoral fellows and tenured staff. However, due to the sample size and incomplete information, we were not able to analyse this formally. Analysing the academic levels more finely, such as incorporating the ‘academic age’ of all conference attendees, would be particularly valuable in ruling in or out this possible explanation for our observed gender bias.

Second, women may tend to be less aware than men of the value of presenting their research comprehensively, or may receive less encouragement or mentorship from supervisors to do so (Sambunjak, Straus & Marušić, 2006). Women may even consider a short amount of visibility on a less comprehensive piece of research as more valuable than less frequent but more comprehensive visibility, while men may value the opposite strategy. There is some evidence in our data to support this latter argument (Figs. 2A and 2B): fewer academic men requested a presentation than women (69% vs. 84%) but when they did so they had a significantly stronger preference for long talks (90% vs. 62%) than academic women.

Third, women have previously been shown to be more risk averse and thus more reticent to publicise their research (Maliniak, Powers & Walter, 2013) particularly if they believed it was at an incomplete stage. If true, this might easily translate into a reluctance to publish thus reducing publication output (a highly valued metric of productivity and often researcher quality) even from an early career stage. Such risk aversion may provide one explanation for the ‘productivity puzzle’ identified across so many fields of academia (Xie & Shauman, 1998; Symonds et al., 2006).

Gender differences in self-perception are apparent in other dimensions of academic output. While a weaker preference for long over short talks, or talks over posters (Isbell, Young & Harcourt, 2012) at conferences and a reduction in the incidence of self-citation might reflect a real difference in how women “play the academic game”, both strategies lead to a reduction in the visibility of women at all stages of their academic career. The broader impact for senior researchers of refusing plenary invitations was comprehensively outlined by Schroeder et al. (2013). Without doubt, under-representation and lower visibility of women scientists at more senior levels may serve as a negative influence on their junior colleagues, perpetuating the tendency for women to leave academia post-PhD. Our data suggests that many of the issues surrounding gender differences in visibility are already present at the junior level and a possible reason for these differences is a lack of awareness of the consequences of under-promoting yourself.

The selection process for the AES conference was not conducted entirely blind to gender (names were known, but gender was unspecified), and we found some evidence (albeit constrained by small sample sizes) that student women were less likely to be assigned a higher visibility long-talk slot than student men. In a recent paper, Knobloch-Westerwick, Glynn & Huge (2013) assessed experimentally the effect of gender on the perceived quality of conference abstracts. Abstracts allocated a ‘male author’ were deemed of greater ‘scientific quality’ than those allocated a ‘female author’. Perhaps more interestingly, was that less than a quarter of the participants assessing the abstracts could actually recall the gender of the abstract authors. This suggests that the observed differences arose through subconscious bias or “processes that the participants were unaware of and did not invest much cognitive capacity in” (Knobloch-Westerwick, Glynn & Huge, 2013).

Future directions

Additional studies are needed to understand whether these results are representative of the broader scientific community. However, our findings contribute to a growing body of literature (Leaper & Robnett, 2011; Schroeder et al., 2013) that raises several interesting questions about why men and women adopt different strategies for presenting their research at conferences. Future studies should seek to tease apart the factors driving this gender difference both in terms of presenters’ choices (e.g., advice from supervisors or colleagues, stage in career, perceived value of presentation, insecurity or risk-aversion), and the selection committee’s choices (e.g. whether the perceived gender of the author or the wording of their abstracts affects talk allocation decisions).

We offer four strategies for future conferences that may help redress the above imbalance. First, greater effort should be made to educate both supervisors and students about innate gender differences in behaviour and how these may exacerbate gender disparity in academia. Historically, most attempts to address gender discrepancies in the workplace have focused on discrimination by others. However, there has been considerably less focus on how gender differences in behaviour can impact professional performance (Schroeder et al., 2013). By explicitly recognizing these behavioural differences, women will be better equipped to evaluate their own decisions, and mentors may be better able to offer advice about how to improve performance and be more competitive. Failure to recognize and educate people about these gender differences makes it more challenging to level the playing field for women and men and narrow the gender gap in academia and other fields of life. In the specific case of conferences, mentoring should highlight that, while it is tempting to view conference presentations as ‘less important’ than perhaps publishing a manuscript, acceptance at a conference may have implications for a future career that extends far beyond the single five- or twelve-minute presentation slot. Second, the language used by conference organisers when inviting submission of abstracts should be reviewed as it may discourage women from submitting abstracts for particular presentation types (Born & Taris, 2010). In particular, requests for talks that present clear and unambiguous results (‘no speculative abstracts’) may bias submissions of longer and more comprehensive talks towards men rather than women, particularly if there is a gender difference in the level of self-doubt about their results. Third, to redress any potential for subconscious gender bias in assessment of conference abstracts, all abstracts (for all conferences) should, at the very least, be scored blind with respect to gender, as is the case for an increasing number of journals (Engqvist & Frommen, 2008). Finally, and admittedly somewhat self-promotionally, to raise awareness of the potential pitfalls associated with reduced visibility and exposure we suggest that societies and particularly conference organisers could provide links to manuscripts and documents that highlight potential gender differences and their implications.

Conclusions

Here we report a striking difference in visibility of men and women at the 2013 Australasian Evolution Society conference. Similar numbers of men and women attended the conference, and there was no gender difference in the decision to present (i.e., submit an abstract). However, women presenters spent on average ∼20% less time presenting their research than men. This discrepancy was driven by gender differences in talk preference: men’s preference for long talks over short talks was stronger than women’s preference. This highlights important gender differences in conference strategy, and merits further attention in order to understand how this may contribute to gender disparities in academia.

While conference abstracts are generally not regarded as research outputs in the field of behavioural and evolutionary biology, conference attendance and presentations (particularly plenary presentations) are important means by which scientists are assessed (Schroeder et al., 2013). Typically, care is taken to reduce the Matilda effect by ensuring equity, if not parity, at the invited plenary level (as indeed was the case for the AES conference). However, if women are less likely to request presentations with a perceived higher value, this is highly problematic. The result is not only a reduction in their immediate visibility but it may have significant implications for future funding opportunities and publication success.

Supplemental Information

Supplemental Information 1 AES Presenter demographics—non identifiable dataset

Click here for additional data file.

We would like to thank Rob Brooks, Lee Ann Rollins, and Ben Fanson for early discussion.

Additional Information and Declarations

Competing Interests

Author Contributions

The authors declare there are no competing interests.

Therésa M. Jones, Rob Lanfear and Matthew R.E. Symonds conceived and designed the experiments, wrote the paper, reviewed drafts of the paper.

Kerry V. Fanson conceived and designed the experiments, wrote the paper, prepared figures and/or tables, reviewed drafts of the paper.

Megan Higgie conceived and designed the experiments, analyzed the data, wrote the paper, prepared figures and/or tables, reviewed drafts of the paper.

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
