# Peer review of "Gender differences in conference presentations: a consequence of self-selection?"

_PeerJ, doi:10.7717/peerj.627_

## Round 0.1 · original submission · Minor Revisions

The reviewers are on the whole positive, regarding the manuscript but make many suggestions regarding interpretation. I would ask that you consider the points raised by the reviewers and include discussion of the issues raised in the manuscript where possible. Please respond as positively as you are able, particularly to the points raised by reviewer two. Although, I do not consider this a major revision requiring re-review, adequate responses to the comments of the reviewers will be pre-requisites for final acceptance. I hope that the peer review process is not too onerous and that on reflection, you feel that the dissemination of your findings is improved.

Reviewer 1 ·

Basic reporting

Clearly written and appropriately set in the context of the relevant literature. This is a self-contained piece of work which might appropriately be regarded as a pilot for a more comprehensive investigation of the issues.

Experimental design

I am not completely convinced that this work falls within the scope of PeerJ. It is reporting what is essentially a social science investigation which happens to have been undertaken using data from an evolutionary biology context. However, given the relevance of the subject matter to equity (which relates to research quality) across the disciplines PeerJ addresses, I would suggest that some elasticity in the scope is reasonable in this case.
The research question is clearly defined and the methods adequately described with data sources explicitly stated.

Validity of the findings

A clearer statement of the conclusions is required. They are currently subsumed into the discussion. Care should be taken to avoid undue reliance on statistical trends; as it stands there is some weight placed on such results.

Additional comments

I would have liked to have seen some suggestions for further work which would explore the relationships between the variables investigated here (preference/allocation) and those identified from other work (e.g. other indicators of representation). I would suggest that it would be worth investigating the rationale of contributers in adopting particular presentations strategies (e.g. are female students advised by supervisors to request only a short talk?).

·

Basic reporting

This submission conforms to the requirements of the journal and is well written.

Experimental design

Understanding the impact of gender on the 'visibility' of research academics presenting at conference is a substantive area for research study and as a question for research is clearly defined. However, the need for this to be investigated in this context is less well explained, it is not clear by whom the research is funded and what where the primary motivations for looking at this within the specific context presented (i.e. the chosen conference)? This should have been more clearly articulated in the introductory paragraphs.
The methods used to address this are straight forward; the stages of the data analysis are clearly defined allowing for future replication. However, no detail is given inrelation to the ethical approval sought for this research or the process for obtaining infomed consent to conduct data analysis on the conference data. This may not have been necessary but this needs to be stated as so.

Validity of the findings

However, the simplicity of the approach could be highlighted as a potential weakness of the paper. First, it is not made clear how the researchers gained access to the data set and what consent they reviewed for re-analysis of this data. Second, the sample size is very small, it is not breoken down into the number of respondent falling into each group i.e. n= XX female students, and the small sample affects the ability for any statistical significance to be inferred from the data and probably contributed to the lack of significance/near significance of the results reported. In order to draw firm conclusions from the results presented the sample size would need to be larger. This could have been a more statistically robust piece of research had the research compared trends in 'visibility' over time (i.e. the results presented here compared against data from previous (same) conferences or against another conference as a comparator). The paper could also be strengthened by making comparisons with other similar conferences in the scientific field. Conference attendance is rightly recognised as only one mode of promoting ones academic work, other means are recognised i.e. through media, justification for the sole focus on conference presentation should be given or else this could have been possitioned as a case study piece- using the example of the once conference?
In addition, the paper rightly identifies the role of academic as a contributory factor in dictating the amount of time that the male/females chose to present for. Nuances within this could have been easily identified by reviewing the demographic composition of the sample and bringing this into the analysis, it is not clear why this was not attempted by this piece of research. This would have added further depth and could presumably have been obtained from data given by conference attendees? Deeper analysis of data and inclusion of additional data to act as a comparator could really have helped this paper to draw more concrete conclusions within the discussion. The conclusions drawn are somewhat ambitious given the issues outlined above. If this was not possible a more detailed justification for the use of data from only this one conference and rationale for small sample size is necessary.
Attention is given to the fact that the papers chosen for presentation were not done so blindly, this would lend one to think that there may have been value in including some qualitative interviewing with the selection panel to understand the motivations behind their selections, this again could have added to the validity of the findings and added an additional dimension to the methodological approach.
The paper could have also been strengthened by the inclusion of recommendations for future research; whilst this is superficially addressed within the discussion a more concrete actionable set of recommendations that possibly highlights some of the key deficiencies of this paper could be included.

·

Basic reporting

The article is clear as to the message that it tries to communicate but I think it would be useful to add information showing how do the proportions of the attending groups reflect gender (im)balance found in the field? The biggest group is Academic Men, which is significantly bigger than Academic Women, and Men Students. The reverse is shown for female attendees. These patterns would be in line with the leaky pipeline phenomenon, but are they?

Lines 141 and 142 contain small typos: 'significance' instead 'significant' and repeated use of "this"

Experimental design

No comments

Validity of the findings

No comments

Additional comments

The article is well structured. The content of the reported study is supported by relevant and recent research discussed in an informative way. The gender issues raised in the specific context of women's participation in science conferences represent an important addition to our understanding of the causes of the 'leaky pipeline' phenomenon and why it persists. This area of gender imbalance as a factor contributing to the accumulation of disadvantages in women's scientific careers, starting from PhD, has been largely overlooked until now. The paper contributes new ways of conceptualising and measuring the effects of gender imbalance in participation, which could lead to establishing better, more gender aware, approaches when designing conference programmes, and create consensus on 'good practice' by adding to the current dominant preoccupation with gender balance among members of committees the prospect of preventing gender bias when selecting speakers and creating equal opportunities for women and men to gain the same visibility within community.
Figure C shows that it is women students who seem to have been most disadvantaged by the selection process. It would be interesting to compare, perhaps in future studies, the language used in the abstracts of the women and men students who were ‘downgraded’ to a short talk or ‘upgraded’ to a long talk by the selection committee. Do women's abstracts contain more “tentative” descriptions of work compared to men's? Another interesting puzzle is the large proportion of Academic Men who did not wish to present, as well as women students, compared to Academic Women and Men Students. Do these groups have different expectations of conference attendance?

---

## Round 0.2 · accepted · Accept

The comments of the reviewers have been adequately addressed and the reporting of the manuscript is sufficiently transparent to facilitate critical appraisal.Thank you for responding positively to the comments of the reviewers.